# Fast abdomen organ and tumor segmentation with nn-UNet

Yajun Wu[1][0000−0002−5636−6859]⋆, Ershuai Wang[1], and Zhenzhou Shao[1]

Department of Research and Development, ShenZhen Yorktal DMIT Co. LTD
`wuyj@yorktal.com`

**Abstract.** The medical imaging community generates a wealth of datasets, many of which are openly accessible and annotated for specific diseases and tasks such as multi-organ or lesion segmentation. However, most datasets are only partially annotated for particular purpose, which hinders the training of multi-talent models. We uses a combination of pseudo labels and partial annotations to generate reliable fully annotated data, avoiding data conflict issues. Then, we designed a fast segmentation method for abdominal organs and tumors based on localization and segmentation. To accelerate inference, we adopt a slice-like downsample for location. To obtain the satisfactory segmentation, we first trained two models for organs and tumors with different target spacing, then combine the results. We also designed a weighted compound loss function and training patches selection strategy to finetuning the model. On the public validation set, the average scores of organ DSC, organ DSC, tumor DSC and tumor NSD are 0.9164, 0.9597, 0.4856 and 0.4221, respectively. Under our development enviroments, the average inference time is 8.54 seconds , the average maximum GPU memory is 4221.49 M, the average area under the GPU memory-time curve is 15074.59. Our code is available at https://github.com/Shenzhen-Yorktal/flare23.

**Keywords:** Segmentation · nnU-Net · Pseudo label.

## 1 Introduction

Abdomen organs are quite common cancer sites, such as colorectal cancer and pancreatic cancer, which are the 2nd and 3rd most common cause of cancer death. Computed Tomography (CT) scanning yields important prognostic information for cancer patients and is a widely used technology for treatment monitoring. In both clinical trials and daily clinical practice, radiologists and clinicians measure the tumor and organ on CT scans based on manual two-dimensional measurements (e.g., Response Evaluation Criteria In Solid Tumors (RECIST) criteria). However, this manual assessment is inherently subjective with considerable inter- and intra-expert variability. Besides, labeling medical images requires professional medical knowledge and rich experience, which makes manual labeling expensive and time consuming. Moreover, existing challenges

---

⋆ Corresponding author

mainly focus on one type of tumor (e.g., liver cancer, kidney cancer). There are still no general and publicly available models for universal abdominal organ and cancer segmentation at present.

Different from existing tumor segmentation challenges, the FLARE2023 focuses on pan-cancer segmentation, which covers various abdominal cancer types. Specifically, the segmentation algorithm should segment 13 organs (liver, spleen, pancreas, right kidney, left kidney, stomach, gallbladder, esophagus, aorta, inferior ven cv, right adrenal gland, left adrenal gland, and duodenum) and one tumor class with all kinds of cancer types (such as liver cancer, kidney cancer, stomach cancer, pancreas cancer, colon cancer) in abdominal CT scans. Also, this challenge provides the largest abdomen training dataset, which includes 2200 3D CT partial labeled scans and 1800 unlabeled scans from 30+ medical centers. However, due to particular clinical purpose at different institutes, these partial labeled scans consists of 219 all organs labeled scans, 484 partial organs labeled scans, 888 only tumor labeled scans and 609 partial organs with tumor scans. What's more, 609 mixed scans only have 5 organs (liver, right kidney, spleen, pancreas, left kidney) and tumors, specifically only 592 of these have the all 5 organs, and the rest 17 scans missing some organ. Besides, the length of scans in axis-z is in range of 74mm to 1983mm which means the region is very different. In a word, the variety of organs, the difference of tumors, the partial annotations and the difference of regions make the segmentation a difficult challenge.

An intuitive strategy is extract each kind of label to make the original partially dataset into 14 binary labeled datasets, then train individual models on each dataset [8]. Afterwards, final segmentation results of all requested organs can be obtained by ensemble the outputs from individual networks. An alternative strategy is to train a single unified model with original partially labeled dataset, where the organs of interest can be segmented simultaneously. In comparison, the latter strategy yields three clear advantages. First, based on the demonstrated benefits of larger training dataset for deep learning models, a unified model trained on union of all partially labeled scans, is anticipated to outperform individual models trained on each binary labeled dataset. Second, during deployment, using a single unified model can lead to faster inference speeds and reduced storage requirements. Lastly, it does not require extra post-processing steps to address conflicting voxel predictions (a voxel being predicted as different classes), a challenge that may arise when using multiple models. Therefore, we adopt the single unified model. To expand training dataset, we also used public pseudo label generated by the best-accuracy-algorithm [17].

Because of significant differences in the scanning area, we adopt common localization followed by segmentation method. Specifically, we first use a light U-Net model to extract abdomen region under large spacing, then segment under fine spacing. To improve tiny organs and tumor segmentation performance, we also proposed a weighted compound loss based on focal loss and dice loss. Besides, we adopt fine-tuning and model ensemble to improve performance further.

## 2  Method

The outline of our method is shown in Fig. 1. Firstly, in order to train unified segmentation model, we generate fully annotated organs-tumor labels, as presented in top Fig. 1. Specifically, we replace pseudo annotation with official label at the same position. Secondly, we train a light model which named ROI extractor to locate the abdomen for reducing computation, resource usage and difference of regions. Besides, we cropped the scans to include abdomen exactly according to the combined labels and trained two segmentation models. Specifically, one named organs model is trained with high resolution, while the other named tumor model is trained with relative low resolution. Both can segment organs and tumors, but their performance are different. As shown in bottom Fig. 1, we ensemble the predictions at the end.

Afterward, we fine-tuning segmentation models to improve tumor and some small organs segmentation performance with particular training patches selection strategy. Specifically, we fine-tuning tumor model only with patches which have tumor at the center. The final prediction is the ensemble of these two models. Last but not the least, all of our models are trained based on nn-UNet [6], which is well known and one of the best baselines for medical image segmentation.

### 2.1  Preprocessing

Image preprocessing is very important for segmentation. Generally, it contains interpolation and normalization. The nn-UNet interpolates isotropic and anisotropic data differently [6]. Median spacing of all training cases is set as default target spacing. For isotropic data, nn-UNet [6] zooms data and segmentation maps with third order spline and nearest-neighbor interpolation respectively. For anisotropic data, nn-UNet [6] zooms data with third order spline in plane first, then interpolates across the out plane axis is done with the nearest interpolation. After that, nn-UNet [6] normalizes CT dataset in a global zero-score manner. Specifically, where a global normalization scheme is determined based on the intensities found in foreground voxels across all training cases.

For ROI extractor, we use a slightly different methods. Specifically, we sample the original data and segmentation maps with a step of 4 in the plane and an integer on the outer axis of plane that makes original spacing close to 5mm. Also, we clipped the data to [-1024, 1024] then normalize it by global mean intensities and standard variance. For segmentation models, we adopt default methods in nn-UNet but change target spacing, [4.0, 1.2, 1.2] for tumor and [2.5, 0.82, 0.82] for organs, respectively.

### 2.2  Proposed Method

**Network:** As mentioned before, we adopt a two-stage segmentation method. The first stage is a ROI extractor, which treat all organs as foreground and the

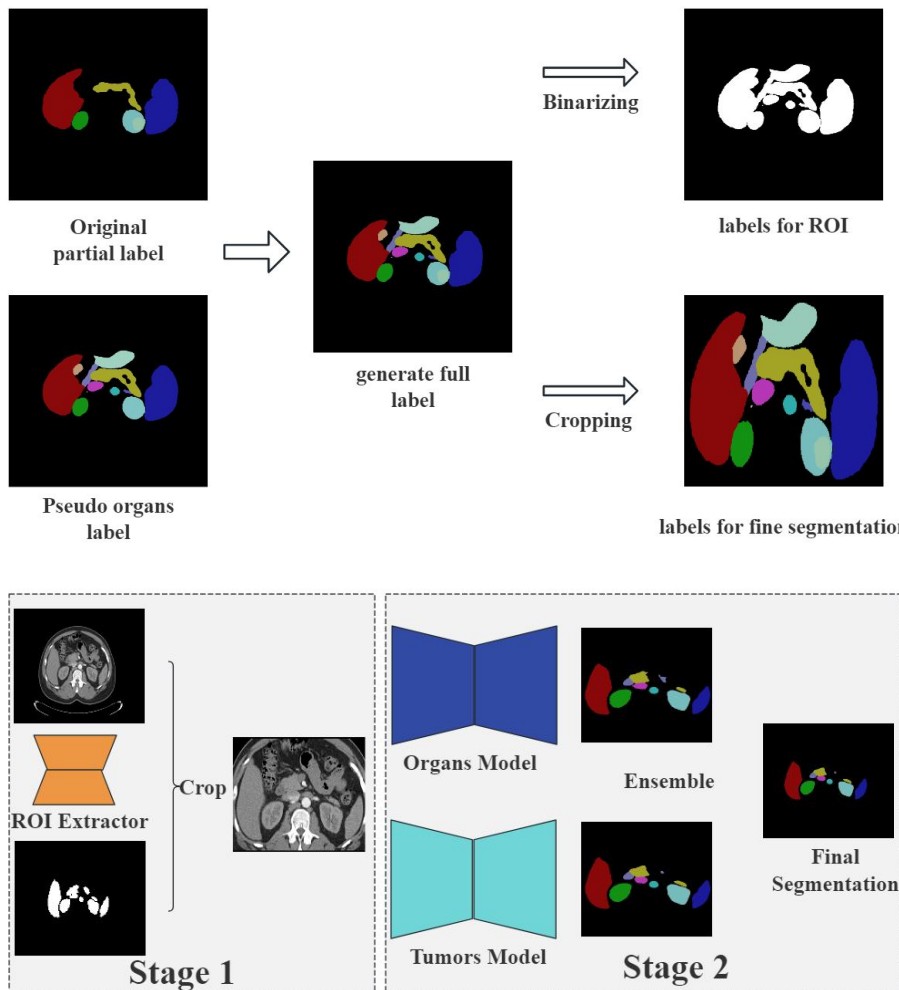

**Fig. 1.** The process of our proposed segmentation framework. (a) shows how we generate the training datasets. (b) shows the final inference pipeline.

others as background. We followed the conventional nn-Unet configuration and got a 5-stages U-Net shown in Fig.2

Specifically, the encoder consists of an initial convolution layer and 5 encoder

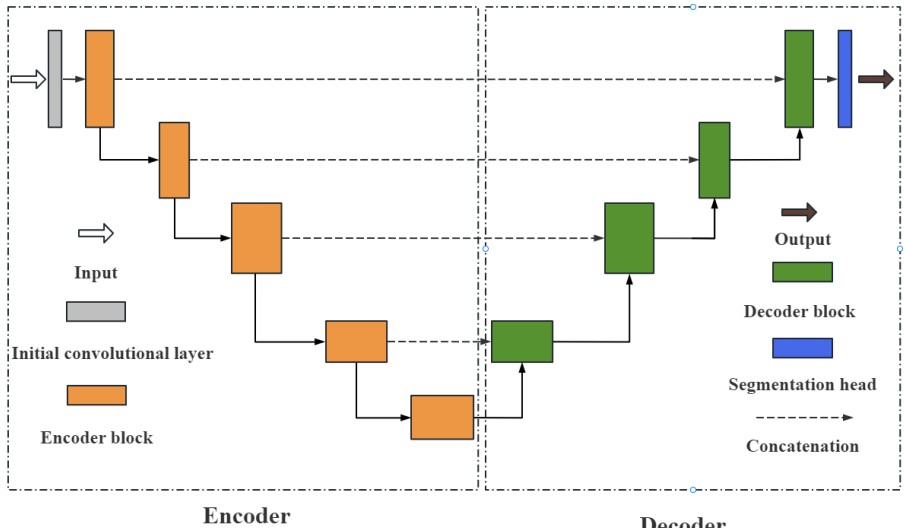

**Fig. 2.** U-Net

blocks with depths of 2, 2, 4, 6, and 4, respectively. Depth denotes number of the sequence of Conv-BatchNorm-ReLU-layers. Strides of the first convolution layer in each encoder block are 1, 2, 2, 2 and 2, respectively. The decoder consists of 4 decoder blocks, each of them consists of a transpose convolution layer which is used to upsample image, and 2 Conv-BatchNorm-ReLU sequential layers which is used to refine features. There are 4 short paths between the encoder and decoder for reusing low level features, enhancing model capacity, and avoiding gradient vanishing. It is worth noting that batch-Norm layer can be absorbed into convolution for acceleration during inference process [3].

Our segmentation models are also based on U-Net, which have different configuration with extractor. Specifically, both models have 6 encoder blocks with depths of 2, 4, 6, 8, 8 and 8, respectively. The basic structure of encoder block is a residual block and the instance normalization is adopted in this model. Decoders are similar to the one in extractor, but with an extra decoder block with stride 2. The only difference is that tumor model has large target spacing while organs model has smaller spacing.

**Loss function and training data selection** Recently, compound losses have been suggested as the most robust losses for medical image segmentation tasks [9]. For model prediction $P$ and label $G$ , we apply the sum of weighted Dice loss [15] and weighted focal loss [7] as the supervised segmentation loss:

$$L = l_d + \lambda l_f \tag{1}$$

$$l_d = -\frac{2\sum_{c=1}^{C} w_c \sum_{b=1}^{B} p_b^c g_b^c}{\sum_{c=1}^{C} \sum_{b=1}^{B} p_b^c + g_b^c} \tag{2}$$

$$l_f = -\sum_{c=1}^{C} \alpha w_c \sum_{b=1}^{B} (1 - p_{t,b}^c)^\gamma \log(p_{t,b}^c) \tag{3}$$

$$p_{t,b}^c = \begin{cases} p_b^c, & g_b^c = 1 \\ 1 - p_b^c & g_b^c = 0 \end{cases} \tag{4}$$

We also proposed a particular data selection strategy. Specifically, we dynamically selected patches during training by the location of organs and tumors. For organs model, we only choose those patches which have organ in the center. For tumors model, we choose patches which have tumor in the center. We believe that this strategy behaves like data oversampling, making the model more focused on specific class.

The loss function together with the data selection improves model performance, especially when we increase weights of small organs and tumors, details are showed in section 4.1.

**Fully annotations:** We generated fully annotated pseudo labels using the best-accuracy-algorithm [17], and then merged them with offical labels of the 2200 partial labeled scans. Specifically, if a voxel has a foreground annotation, we replace the pseudo label with this annotation, otherwise we use generated pseudo label as the ground-truth. The 1800 unlabeled images were not used because of missing information of tumors.

**Strategies to improve inference speed and reduce resource consumption**

We improve inference from four aspects.

- **Interpolate the probability by GPU:** We found that restore the shape of predicted probability to the original shape was the most time-consuming. The reason behind this is that the computation is too huge for third order spline interpolation, especially interpolate by CPU. Therefore, we utilize the powerful parallel computing capabilities of GPU to reduce running time. Considering that computational load and memory usage increase with the number of target channels and volumes size, we adopted a block calculation method for large CT scans to avoid GPU memory overflow. Specifically, we restore probability every 150 slices each time.
- **Generate labels by GPU:** Default label generation method of nn-UNet is implemented on CPU. We use GPU again to make it faster.
- **Replace preprocessing by torch:** Default preprocessing is implemented by the toolkit for sciPy, which is slightly slower than torch. To reduce running time further, we decide to adopt interpolation of torch instead.

– **MultiThreading:** Another phenomenon is that even if the ROI model is smaller and there is less ROI data, ROI extraction expands much longer time than fine segmentation. We believe this is due to initialization of libraries of pytorch. To solve this problem, we use multithreading method to load models and complete initialization process when reading and preprocessing images.

Using these tricks, we were ultimately able to segment almost all of the validation cases within 15 seconds. Due to precise ROI and interpolate the large images by block, 4GB GPU Memory is enough. Therefore, we did not make other changes for the GPU memory resource consumption.

### 2.3 Post-processing

Inspired by anatomy and the uniqueness of organs, the largest connected component-based post-processing is commonly used in medical image segmentation. In this work, we found that keep the largest connectivity component can improve the DSC scores of liver, spleen, pancreas, LAG, RAG and stomach. Due to the variety of tumors, retaining the largest connectivity component severely decrease the tumor DSC score. Therefore, we only keep the largest connected component for liver, spleen, pancreas, LAG, RAG and stomach, while keep the others unchanged.

## 3 Experiments

### 3.1 Dataset and evaluation measures

The FLARE 2023 challenge is an extension of the FLARE 2021-2022 [11][12], aiming to aim to promote the development of foundation models in abdominal disease analysis. The segmentation targets cover 13 organs and various abdominal lesions. The training dataset is curated from more than 30 medical centers under the license permission, including TCIA [2], LiTS [1], MSD [16], KiTS [4,5], and AbdomenCT-1K [13]. The training set includes 4000 abdomen CT scans where 2200 CT scans with partial labels and 1800 CT scans without labels. The validation and testing sets include 100 and 400 CT scans, respectively, which cover various abdominal cancer types, such as liver cancer, kidney cancer, pancreas cancer, colon cancer, gastric cancer, and so on. The organ annotation process used ITK-SNAP [18], nnU-Net [6], and MedSAM [10].

It is noticeable that we selected our training dataset for fine segmentation. Specifically, our training dataset only includes scans which have tumors and the tumor's bounding box must located in the bounding box of the organs.

The evaluation metrics encompass two accuracy measures—Dice Similarity Coefficient (DSC) and Normalized Surface Dice (NSD)—alongside two efficiency measures—running time and area under the GPU memory-time curve. These metrics collectively contribute to the ranking computation. Furthermore, the

running time and GPU memory consumption are considered within tolerances of 15 seconds and 4 GB, respectively.

### 3.2   Implementation details

**Environment settings** The development environments and requirements are presented in Table 1.

**Table 1.** Development environments and requirements.

| | |
|---|---|
| Windows/Ubuntu version | Windows 10 pro |
| CPU | Intel(R) Core(TM) i7-10700kF CPU@3.80GHz |
| RAM | 16×4GB; 2.67MT/s |
| GPU (number and type) | One NVIDIA RTX 3090 24G |
| CUDA version | 11.1 |
| Programming language | Python 3.8 |
| Deep learning framework | Pytorch (Torch 1.10, torchvision 0.9.1) |
| Link to code | |

**Training protocols** The training protocols of ROI extractor and fine segmentation model are listed in Table 2 and Table 3. We adopt data augmentation of additive brightness, gamma, rotation, mirroring, scaling and elastic deformation on the fly during training.

During training process, the batch size is 2 and 250 batches are randomly selected from the training set per epoch, the patch size is fixed as 32 * 128 * 192. For optimization, we train it for 2000 epochs using SGD with a learning rate of 0.01 and a momentum of 0.95. Besides, the learning rate is decayed following the poly learning rate policy. As for fine-tuning, we reduce the initial learning rate to 0.0001 and increase the batch size to 4.

## 4   Results and discussion

### 4.1   Quantitative results on validation set

At the very beginning, we trained a organ-only segmentation model using the 219 scans with fully organs annotation. The model was tested on public validation set which got 0.8715 average DSC score and 0.9342 NSD score. Due to limited training data and the presence of tumors, it is lower than the winner of FLARE2022.

Next, we constructed a larger training set with 2200 scans as described in section 2 and trained a basic segmentation model. Benefit from more training data, the organ DSC, NSD, tumor DSC, and NSD scores of this model are 0.9078,

**Table 2.** Training protocols for the ROI model.

| | |
|---|---|
| Network initialization | "he" normal initialization |
| Batch size | 2 |
| Patch size | 80×128×128 |
| Total epochs | 2000 |
| Optimizer | SGD with nesterov momentum ($\mu = 0.95$) |
| Initial learning rate (lr) | 0.01 |
| Lr decay schedule | poly learning rate policy $lr = 0.01 * (1 - \frac{e}{m})^2$ |
| Training time | 76.5 hours |
| Number of model parameters | 10.66M |
| Number of flops | 103.36G |

**Table 3.** Training protocols for the refine model.

| | |
|---|---|
| Network initialization | "he" normal initialization |
| Batch size | 2 or 4(fine-tuning) |
| Patch size | 32×128×192 |
| Total epochs | 2000 |
| Optimizer | SGD with nesterov momentum ($\mu = 0.95$) |
| Initial learning rate (lr) | 0.01 or 0.0001(fine-tuning) |
| Lr decay schedule | poly learning rate policy $lr = 0.01 * (1 - \frac{e}{m})^2$ |
| Training time | 104.5 hours |
| Number of model parameters | 68.34M |
| Number of flops | 158.13G |

0.9461, 0.3841 and 0.2994, respectively. Then, we fine-tuned the model. Specifically, we increase the batch size to 4, set the initial learning rate to 0.0001, selected the patches with organs in the center and retrained the model for 1000 epochs. This increase the organs DSC and NSD scores to 0.9153 and 0.9558, which are 0.0075 and 0.0097 higher. Besides, the tumors DSC and NSD scores are 0.3886 and 0.3237, which are also slightly better.

Obviously, there are significant differences in segmentation performance between organs and tumors. We argue that it is caused by the variety of tumors, imprecise annotation of tumor especially of tumor boundaries and the data imbalance between organs and tumors. In order to improve performance of tumors, we increase the loss weights of tumors, increase the target spacing to [4.0, 1.2, 1.2] and retrain the model from scratch. As expected, the tumor DSC score and NSD score improved to 0.4426 and 0.3732, respectively, which were 0.0585 and 0.0738 higher than the organ model. Then, we also fine-tuned it with patches that have tumors in the center. This further improves the model, resulting in tumor DSC and tumor NSD scores of 0.4803 and 0.4201, respectively. Besides, this model got the organs DSC and NSD scores of 0.9055 and 0.9537, respectively.

To achieve better segmentation performance, we ensemble these two models. Specifically, if the prediction of the tumor model is a tumor, then the final label is a tumor, otherwise the final label is the prediction of the organ model, as showed in equation 5. Our final validation scores of organ DSC, organ NSD, tumor DSC and tumor NSD are 0.9165, 0.9597, 0.4803 and 0.4201 respectively, details are presented in Table 4.

$$L(x_i) = \begin{cases} \text{Tumor}, & M_{tumor}(x_i) = \text{Tumor} \\ M_{organ}(x_i), & else \end{cases} \tag{5}$$

### 4.2   Qualitative results on validation set

Fig.3 presents some well-segmented cases in the public validation set. Similar to the DSC scores, there is little visual difference in the segmentation of liver, spleen, and aorta compared to the ground truth. We believe this is due to the intensity homogeneity, clear boundaries and good contrast. On the contrary, Fig.4 presents some challenging cases, there are apparent difference in the segmentation of pancreas, gallbladder, duodenum, esophagus and tumors. We believe this is due to the smaller size, unclear boundaries and the heterogeneity, especially for tumors.

### 4.3   Results on final testing set

The FLARE23 organizer collected 400 CT scans from several center sites as the final testing set. Our final performance on this hidden testing set are presented in Table 4. We can see the average organs DSC score, organs NSD score, tumors DSC score and NSD score are 0.9211, 0.9589, 0.6467 and 0.5432, respectively, which are very close to the scores on validation set. This means our models have good robustness.

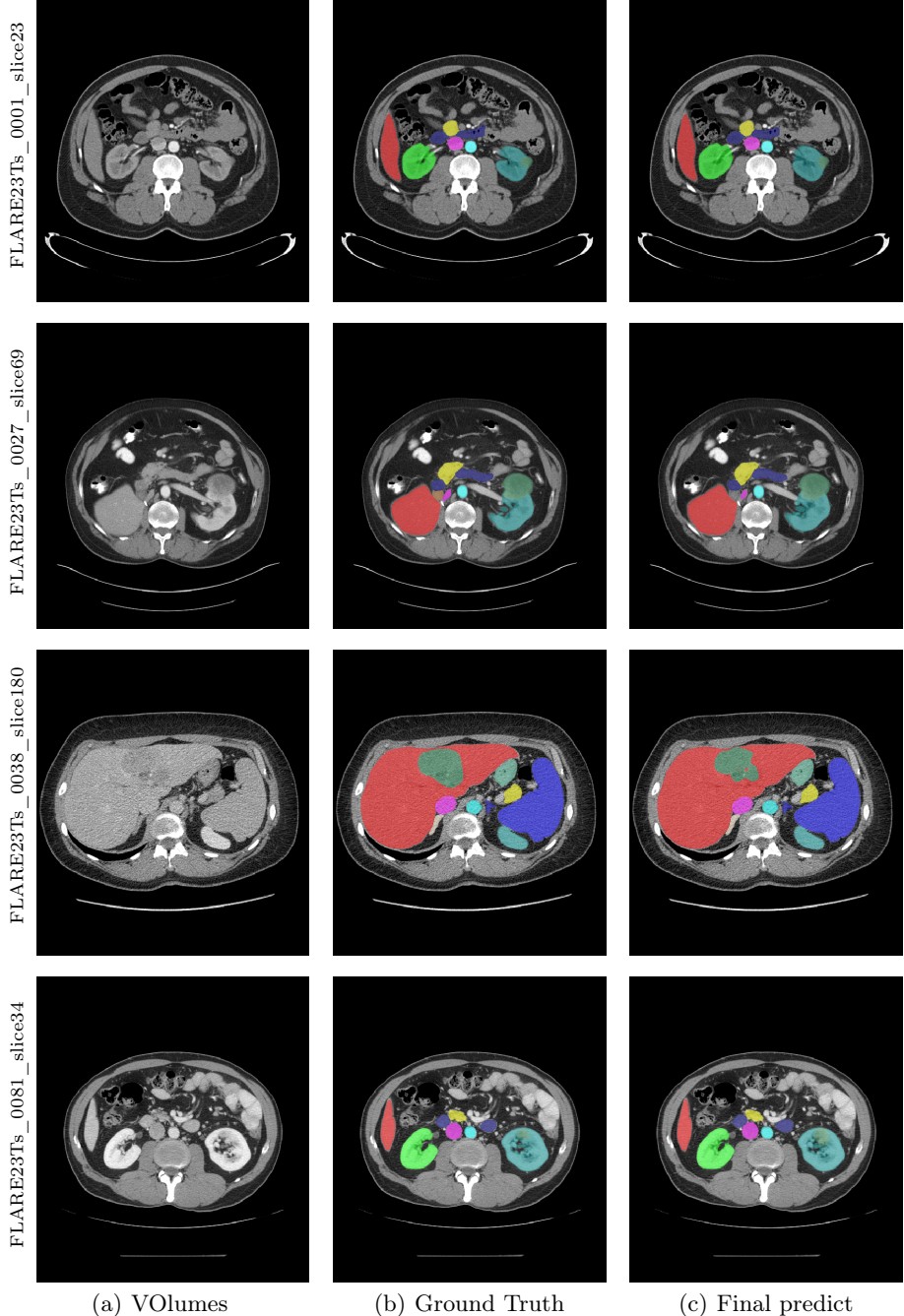

(a) VOlumes          (b) Ground Truth          (c) Final predict

**Fig. 3.** Well-segmented cases.

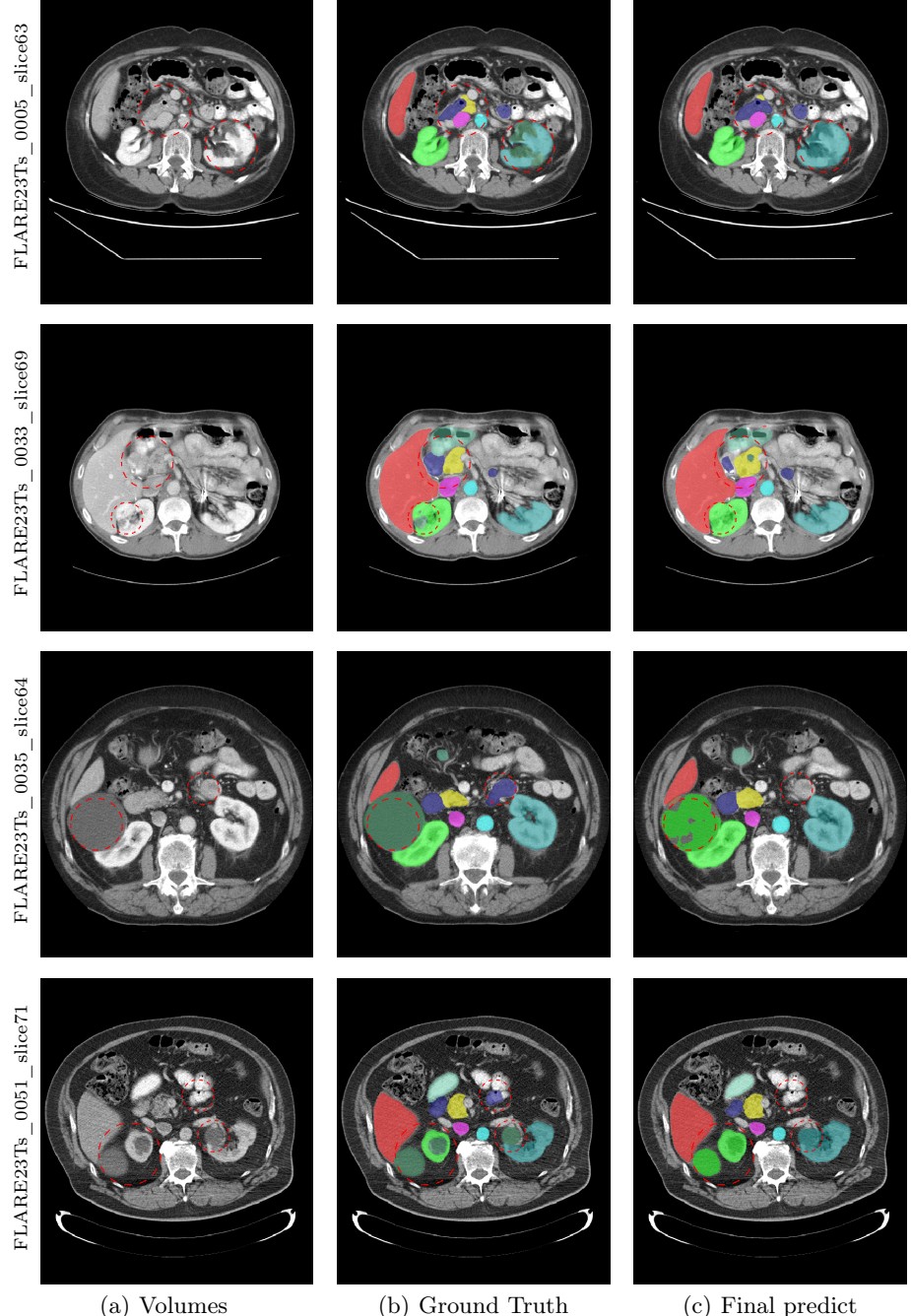

(a) Volumes          (b) Ground Truth          (c) Final predict

**Fig. 4.** Challenging cases. The red dotted circles have significant difference between ground-truth and prediction.

**Table 4.** Quantitative evaluation results

| Target | Public Validation | | Online Validation | | Testing | |
|---|---|---|---|---|---|---|
| | DSC(%) | NSD(%) | DSC(%) | NSD(%) | DSC(%) | NSD (%) |
| Liver | 98.40 ± 0.44 | 99.26 ± 0.73 | 98.16 | 99.10 | 97.23 | 98.13 |
| Right Kidney | 95.07 ±7.45 | 96.25±7.88 | 94.97 | 95.69 | 95.10 | 95.15 |
| Spleen | 98.24±0.91 | 99.25±1.30 | 98.19 | 99.30 | 97.91 | 99.07 |
| Pancreas | 87.93±7.96 | 97.31±4.09 | 87.28 | 96.72 | 90.74 | 97.11 |
| Aorta | 97.35±1.84 | 99.22±2.15 | 97.23 | 99.26 | 97.76 | 99.75 |
| Inferior vena cava | 93.30±4.30 | 94.02±4.60 | 92.11 | 92.43 | 92.96 | 93.86 |
| Right adrenal gland | 89.98±4.25 | 98.22±2.04 | 89.06 | 97.87 | 87.52 | 96.27 |
| Left adrenal gland | 89.03±5.13 | 97.61±2.84 | 87.90 | 96.45 | 89.40 | 96.67 |
| Gallbladder | 85.80±23.77 | 86.58±24.63 | 88.29 | 89.58 | 83.96 | 86.24 |
| Esophagus | 84.74±15.70 | 93.88±13.45 | 84.78 | 94.02 | 89.55 | 97.24 |
| Stomach | 94.36±3.51 | 96.89±4.16 | 94.73 | 97.37 | 94.99 | 97.38 |
| Duodenum | 83.76±9.05 | 94.38±6.46 | 83.56 | 94.48 | 88.01 | 96.11 |
| Left kidney | 95.61±5.77 | 94.98±8.69 | 95.01 | 95.38 | 94.32 | 94.86 |
| Tumor | 53.63±37.10 | 47.82±34.28 | 48.56 | 42.21 | 64.67 | 54.32 |
| Average Organs | 91.81±5.21 | 95.99±3.44 | 91.64±5.10 | 95.97±2.86 | 92.11±4.42 | 95.89±3.36 |

## 4.4 Efficiency analysis

As described in section 2.2, we speed up the inference on four aspects, the inference running time details are present in Table 5. We can see that interpolating the probability to the original image size with GPU reduce the average running time by 23.88 seconds, generating the labels with GPU reduce the average running time by 2.88 seconds, preprocessing the image with torch reduce the average running time by 1.23 seconds and adopt multithreading reduce another 0.85 seconds.

Despite of using models ensemble, our method consumes less than 4GB GPU memory and can segment most of the scans within 15 seconds on official testing environment, except the extremely large cases.

**Table 5.** Comparison of accelerate strategies on local machine. Basic denotes inference with default nn-UNet. S1 denotes interpolating probability to original size with GPU. S2 denotes generating labels with GPU. S3 denotes preprocessing image with torch instead. S4 denotes multithreading, which means load models while reading and preprocessing images.

| Strategies | Average Running Time (s) | Max Running Time (s) | Average GPU (MB) | Max GPU (MB) |
|---|---|---|---|---|
| Basic | 37.14 | 95.35 | 4195 | 4624 |
| +S1 | 13.26 | 25.41 | 4205 | 4625 |
| +S2 | 10.68 | 18.41 | 4197 | 4624 |
| +S3 | 9.45 | 14.14 | 4203 | 4624 |
| +S4 | 8.60 | 11.74 | 4226 | 4721 |

**Table 6.** Quantitative evaluation of segmentation efficiency in terms of the running time and GPU memory consumption. Total GPU denotes the area under GPU Memory-Time curve. Evaluation GPU platform: NVIDIA QUADRO RTX5000 (16G).

| Case ID | Image Size | Running Time (s) | Max GPU (MB) | Total GPU (MB) |
|---|---|---|---|---|
| 0001 | (512, 512, 55) | 14.18 | 3674 | 17980 |
| 0051 | (512, 512, 100) | 9.75 | 3872 | 18538 |
| 0017 | (512, 512, 150) | 10.45 | 3816 | 16828 |
| 0019 | (512, 512, 215) | 9.08 | 3750 | 14843 |
| 0099 | (512, 512, 334) | 9.19 | 3426 | 15285 |
| 0063 | (512, 512, 448) | 10.81 | 3548 | 18534 |
| 0048 | (512, 512, 499) | 11.45 | 3638 | 20567 |
| 0029 | (512, 512, 554) | 14.05 | 3928 | 28659 |

### 4.5   Limitation and future work

As showed in section 4.1, the DSC scores of gallbladder, esophagus, duodenum and pancreas are lower than 0.9, the DSC and NSD scores of tumor are much lower than organs. Besides, our final results are fused by two similar refine models, which is slightly time-consuming and complex. In the future, we will focus on improving the segmentation of small organs and tumors. Also, knowledge distillation maybe adopted to combine the two models into one.

## 5   Conclusion

In this work, we adopt pseudo labels to address the conflict of background and missing annotations. Then we proposed a weighted compound loss and a particular training-patches selection strategy to alleviate the class imbalance problem. Finally, we improve the performance by fine-tuning and model ensemble. These techniques may be helpful for other medical image segmentation tasks.

**Acknowledgements** The authors of this paper declare that the segmentation method they implemented for participation in the FLARE 2023 challenge has not used any pre-trained models nor additional datasets other than those provided by the organizers. The proposed solution is fully automatic without any manual intervention. We thank all the data owners for making the CT scans publicly available and CodaLab [14] for hosting the challenge platform.

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

**Table 7.** Checklist Table. Please fill out this checklist table in the answer column.

| Requirements | Answer |
| --- | --- |
| A meaningful title | Yes |
| The number of authors ($\leq 6$) | 3 |
| Author affiliations, Email, and ORCID | Yes |
| Corresponding author is marked | Yes |
| Validation scores are presented in the abstract | Yes |
| Introduction includes at least three parts: background, related work, and motivation | Yes |
| A pipeline/network figure is provided | Figure 1 |
| Pre-processing | Page 3 |
| Strategies to use the partial label | Page 3,5 |
| Strategies to use the unlabeled images. | Page 5 |
| Strategies to improve model inference | Page 6 |
| Post-processing | Page 6 |
| Dataset and evaluation metric section is presented | Page 6, 7 |
| Environment setting table is provided | Table 1 |
| Training protocol table is provided | Table 2,3 |
| Ablation study | Page 9 |
| Efficiency evaluation results are provided | Table 5, 6 |
| Visualized segmentation example is provided | Figure 3 |
| Limitation and future work are presented | Yes |
| Reference format is consistent. | Yes |