# OpenReview forum: "Fast abdomen organ and tumor segmentation with nn-UNet"
_MICCAI.org/2023/FLARE — Submitted to FLARE 2023_

### Official Review · Reviewer_9edC · 2023-09-19
**Quick and concise**

**Rating:** 9
**Confidence:** 4

**Review:**

Pros:

1.The article has a complete structure.

2.The proposed method can greatly reduce inference time

Cons:

1.The explanation of a separate decoder for tumors is not clear enough and is not reflected in the structure figure, which does not allow people to intuitively understand how tumors are specially processed.

---

> ### Author Response · Authors · 2023-11-03
>
> Dear reviewer,
> Thank you for your comments and kindly advice.
> Actually, there are two seperate networks for fine segmentation, then we ensemble the prediction.
> I will modify the Fig 2 to make the whole pipeline clear and explain the reason of using seperate models in section 2.

---

> > ### Comment · Reviewer_9edC · 2023-11-20
> > **Supplement**
> >
> > The text in the image cannot be selected. It is recommended to use PDF format when importing the project

---

### Official Review · Reviewer_S1d9 · 2023-09-22
**Fast abdomen organ and tumor segmentation with nn-UNet**

**Rating:** 7
**Confidence:** 5

**Review:**

The authors design a weighted compound loss function and a training patches selection strategy for finetuning the model. These techniques contribute to improving the accuracy and robustness of the segmentation results.

---

> ### Author Response · Authors · 2023-11-03
>
> Dear reviewer,
> Thank you for your kindly comments!

---

### Official Review · Reviewer_UvSN · 2023-10-04
**This paper is very clear in details and experiment settings**

**Rating:** 9
**Confidence:** 5

**Review:**

As a challenge manuscript, this paper includes detailed experiment settings, model complexities, parameters, training details, and evaluation approaches.

---

> ### Author Response · Authors · 2023-11-03
>
> Dear reviewer,
> Thank you for your kindly comment!
> We make a better Fig2 which is helpful for understanding the whole pipeline!
> Thank you again.

---

### Official Review · Reviewer_55Ph · 2023-10-21
**Fast abdomen organ and tumor segmentation with nn-UNet**

**Rating:** 8
**Confidence:** 4

**Review:**

Pros:

- Comprehensive and well-structured presentation of experiments and methodology.
- Effective techniques, including a weighted compound loss function and training patch selection, leading to improved segmentation accuracy.
- Significant reduction in inference time with the proposed method.

Cons:

- Lack of clarity in explaining the separate decoder for tumors, making it difficult to understand its specific role.
- The structure figure does not intuitively depict how tumors are processed, potentially causing confusion among readers.

---

> ### Author Response · Authors · 2023-11-03
>
> Dear reviewer,
> Thank you for your comments and kindly advice.
> Actually, there are two seperate networks for fine segmentation, then we ensemble the prediction.
> I will modify the Fig 2 to make the whole pipeline clear and explain the reason of using seperate models in section 2.

---

### Official Review · Reviewer_4qKy · 2023-10-23
**A nnUNet that enables accurate and fast segmentation**

**Rating:** 8
**Confidence:** 5

**Review:**

The method proposed in this paper accurately and rapidly accomplishes organ tumor segmentation in each case. The paper clearly spells out the complete process as officially required. However, the following problems exist:
- The paper proposes a two-stage method, but Figure 2 has only one stage?

---

> ### Author Response · Authors · 2023-11-03
>
> Dear reviewer,
> Thank you very much for your comments. I will update the Fig2 to make the whole pipeline clear.
> Ps: We locate the abdomen by the ROI extractor which is the network at rop right of Fig2; then segment the organs and tumors by 2 fine models(tumor model and organ model) at the bottom right of Fig2.

---

### Decision · Program_Chairs · 2023-10-24

Accept